# When Preschoolers Interact with an Educational Robot, Does Robot Feedback Influence Engagement?

**Mirjam de Haas** [1,2,*] **, Paul Vogt** [1,3] **and Emiel Krahmer** [2,4]

1 Department of Cognitive Science and Artificial Intelligence, Tilburg School of Humanities and Digital Sciences, Tilburg University, 5037 AB Tilburg, The Netherlands; p.a.vogt@pl.hanze.nl
2 Tilburg Center for Cognition and Communication, Tilburg University, 5037 AB Tilburg, The Netherlands; e.j.krahmer@tilburguniversity.edu
3 School of Communication, Media & IT, Hanze University of Applied Sciences, 9747 AS Groningen, The Netherlands
4 Department of Communication and Cognition, Tilburg University, 5037 AB Tilburg, The Netherlands
* Correspondence: mirjam.dehaas@tilburguniversity.edu

**Abstract:** In this paper, we examine to what degree children of 3–4 years old engage with a task and with a social robot during a second-language tutoring lesson. We specifically investigated whether children's task engagement and robot engagement were influenced by three different feedback types by the robot: adult-like feedback, peer-like feedback and no feedback. Additionally, we investigated the relation between children's eye gaze fixations and their task engagement and robot engagement. Fifty-eight Dutch children participated in an English counting task with a social robot and physical blocks. We found that, overall, children in the three conditions showed similar task engagement and robot engagement; however, within each condition, they showed large individual differences. Additionally, regression analyses revealed that there is a relation between children's eye-gaze direction and engagement. Our findings showed that although eye gaze plays a significant role in measuring engagement and can be used to model children's task engagement and robot engagement, it does not account for the full concept and engagement still comprises more than just eye gaze.

**Keywords:** child–robot interaction; engagement; second-language learning; robot tutor; preschool children

## 1. Introduction

In recent years, the interest in using robots for educational purposes has increased substantially [1,2] due to the growing numbers of students in classrooms, shrinking school budgets and the fact that robots can possibly exhibit social behaviors that can benefit children's learning [1]. One application in the educational domain that utilizes robots is second-language (L2) learning [2,3], in which robots are often used as tutors to support children's L2 acquisition. In order to be successful as a robot tutor, the robot should be able to engage the children in order to motivate them during the task. The aim of this article is to investigate children's engagement during a second-language tutoring lesson with a social robot.

Engagement plays an important role in learning. Engaged children are more motivated and are more likely to continue longer with their learning tasks than disengaged children. The more time children are actively interacting with a certain task, the more children can learn from that task. The engagement of elementary and middle school children has frequently been studied, being linked numerous times to children's academic performances, e.g., [4,5].

In human–robot interactions (HRI), it is common for people to initially be highly engaged but quickly start to become less engaged as the task continues due to its repetitive

nature and the novelty of the task wearing off. This novelty effect is observed in both the engagement with the robot as a social partner (robot engagement) and in the engagement with the task within the robot interaction (task engagement) [6]. This distinction between task engagement and robot engagement is important because children can be engaged with the learning task in front of them, but not with the tutor, or vice versa. Both engagement types can have an influence on children's learning [6,7], although the results are inconsistent [7,8]. Previous studies on HRI typically only measured the engagement with the robot and not with the task given to the participant [6,9,10]. The reason for this is that researchers are specifically interested in the effect of their manipulation, which is often a result of the robot's behavior. However, it is also worth examining task engagement [11], because this may reveal whether the learner's engagement decreased in response to the experimental task or the robot's behavior.

There are several methods that are able to stimulate and maintain children's task engagement and robot engagement, and one of them is feedback. Providing children with the correct form of feedback is essential, as different children seem to respond better to different feedback types [12]. On the one hand, positive feedback can motivate children, keep them engaged during a task and can activate their learning behavior [13]. On the other hand, for other children, it might decrease their performance, when the children receive this feedback too often, it becomes too repetitive and, as a result, the children become less engaged [14]. Children can also respond differently to negative feedback, especially young children. As young children (preschoolers) quickly absorb all the information around them and rely on their environment for (correct) input, they tend to benefit more when they receive corrective feedback than adults would [15]. In contrast, younger children might be more sensitive to explicit negative feedback than older children, particularly when it guides them to notice errors [16–18]. Moreover, negative feedback can lead to frustration which can decrease children's motivation to fully participate in the task and therefore decrease children's task engagement [19].

This article aims to investigate children's task engagement and robot engagement during a second-language (L2) learning task with a robot by specifically focusing on the role of the robot's feedback on the children's engagement. Moreover, we investigate the role of children's eye gaze on their task engagement and robot engagement. In the following sections, we provide an overview of earlier work on engagement in child–robot interactions and feedback in education. We then explain the design of the experimental study and, finally, we will present the results and discuss our findings.

*1.1. Background*

1.1.1. Engagement

Numerous studies across the HRI field focus on engagement. After all, the key to continuing to use robots in different fields is when people remain interested in robots, especially over time. For many people, robots are something new and hence interesting. However, over time, this interest may change. Consequently, engagement is widely studied and frequently, when researchers refer to the concept of engagement, a variety of definitions are used. The most commonly used definition in HRI is by Sidner et al. [20], who defined engagement as "the process by which individuals in an interaction start, maintain and end their perceived connection to one another" (page 141), but there are others who argue that it is more than a cognitive process and explain engagement as a multidimensional concept of a cognitive dimension (such as attention), an affective dimension (such as emotions) and a behavioral dimension (such as the execution of tasks) [21,22]. Although there has been a large variation in the definition of engagement and in how it has been studied, there is an agreement that engagement is a multidimensional concept.

Children are normally very engaged with the robot, but this quickly decreases over time which has been shown in numerous experiments, e.g., [23–25]. It is, therefore, important to understand which robot behavior can lead to a positive effect on children's engagement. Many studies have investigated the effect of robot behavior on children's

engagement, looking at different robot behaviors such as the robot's gestures [10], expressiveness of the voice [26], or the role of the robot [11,27]. De Wit et al. [10] investigated the effect of gestures on 5-year-old children's engagement and found positive effects. Kory-Westlund et al. [26] found that 5-year-old children were more engaged with a robot exhibiting expressive behaviors. A recent study showed that 5- to-7-year old children who interacted with a robot acting as a peer showed more affect during the interaction than when interacting with a robot acting like a tutor [27].

A disadvantage of these studies is that they focus on engagement with the robot instead of the task. However, it is possible that task engagement is more important for learning. A follow up study by de Wit et al. [28] found a positive effect of robot gestures on children's robot engagement but not on task engagement nor learning gain. Moreover, Zaga et al. [11] investigated task engagement during a robot tutoring lesson. In their experiment, they compared a robot behaving as a peer and a tutor and found that children were more engaged in the task and solved the task faster with the peer-like robot than with the tutor-like robot.

Similarly to how there are different definitions of engagement, there are also different methods for measuring engagement [6]. For adults, questionnaires can be used as self-reported measures. This can be useful to determine participants' own reflection of the interaction. Unfortunately, questionnaires only provide a total rating after the interaction and not during the interaction, and are difficult to use with children. Other methods are based on video or audio data and measure participants' output behaviors, such eye gaze, head movements (nodding), verbal utterances and facial expressions or a combination of these behaviors [29–31]. Eye gaze is especially important, because it can indicate where the participant's attention is directed and can relatively easily be measured automatically, making it ideal for real-time engagement tutoring interactions.

Some studies have examined the role of eye gaze within engagement [30–32]. Nakano and Ishii [32] and Ishii et al. [31] used automatic gaze direction to initiate the asking of probing questions by the robot whenever the participant looked away from the robot, indicating disengagement. This showed to have a positive effect on the participants' non-verbal and verbal behaviors. However, they concentrated their study on the social interaction between the robot and participant and did not investigate what happens when a task is in front of the participant. This can result in different eye-gaze behaviors such as looking away from the robot more often. Rich et al. [30] combined mutual gaze and joint attention to determine the participant's engagement and this combination increased the participant's attention to the robot. However, these studies do not differentiate between robot engagement and task engagement and it is possible that they actually measured participants' engagement with the robot. Moreover, these studies did not investigate whether it is feasible to monitor eye gaze with children and whether children's eye gaze relates to engagement. Although eye gaze only focuses on one aspect of engagement, it undoubtedly plays a role because it can show the direction of the participant's attention which is one of the three dimensions of engagement according to [21]. It does not, however, explain the whole concept. Therefore, it would be interesting to examine how large the role of this single element is and whether this role is large enough to successfully predict children's task engagement and robot engagement during a L2 learning tutoring session.

### 1.1.2. Feedback

Research on second-language learning has demonstrated the importance of feedback and engagement in children's language learning performance in human–human studies [33]. While the role of feedback has extensively been studied in human–human interactions, in the field of child–robot interaction it is largely understudied (see [7,9]). In order to design social robots as effective L2 tutors, it is therefore important to investigate how a social robot should provide feedback to optimize children's engagement.

In general, educational robots are designed based on how human teachers interact with their pupils; however, in classroom settings, children not only receive feedback from their

teachers but also from their peers. Teachers normally provide a combination of positive and negative feedback. They use explicit positive feedback to encourage the children and they recast the children's answers to provide corrections as a type of implicit negative feedback [34]. Their positive feedback can result in children becoming more engaged with the task and when they are fully engaged, they learn faster and continue longer with the task [24,35,36].

The use of recasts during L2 learning provides a subtle way to correct the children's mistakes. In the case of a recast, the adult will repeat the utterance, but rephrase the incorrect part into a correct one (e.g., when a child had said: "The cat is jumping", he/she may be corrected by the adult's utterance: "Ah right, the *dog* is jumping"). The use of recasts is additionally intended to avoid providing demotivating comments found in explicitly negative feedback.

Children do not only receive feedback from their teachers, they also receive feedback from peers in their classroom [37]. In contrast to the implicit feedback that adults provide, children tend to use more explicit language ("No, you are wrong!") [15]. It has been argued that explicit feedback can have a more substantial impact on learning than implicit feedback [38]. However, the potential side effect of providing explicit feedback is that children's engagement decreases. As shown, all of these different forms of feedback provide children with the correct information but in a different manner, and consequently, these different forms may have a different influence on the children's engagement. In addition, children might have feedback preferences, where one child might remain more engaged with explicit feedback, while implicit feedback might stimulate engagement more for another child.

Given that learners do not exclusively receive feedback from adults, the design of robot feedback on the basis of the teacher's feedback might not always be the most optimal for children's development. For example, research has shown that the presence of a peer improves learning potential [37], and as a result, some researchers have argued that educational robots might work better when presented as peers, especially since children may treat the robot as a peer rather than a teacher in long-term interactions [24]. Therefore, it might be better to design feedback provided by a robot based on children's peer interactions.

The use of feedback in child–robot interaction studies has not been extensively studied. Adult–robot interaction studies showed that participants listened more to negative feedback provided by a robot than negative feedback provided by computers [39], participants learned more words during an L2 learning task from a robot providing only negative feedback than a robot providing only positive or no feedback [40], and positive feedback positively influenced adults' acceptance of the robot as an instructor [41] and increased adults' motivation [39]. However, it is difficult to relate these results to children because children learn differently to adults.

In child–robot interactions, most studies report the use of praise or various types of negative feedback, such as introducing a doubt ("Are you sure?") instead of negative feedback [42], providing hints ("I think it was the other one") [43] or providing children with an extra attempt after an incorrect answer [10], but these studies did not investigate the effect of these feedback utterances on children's learning gain or engagement. Only two studies have investigated the use of robot feedback in language learning [7,9]. In a study by Ahmad et al. [9], children of eight to ten years old played a game with a robot on a tablet. The robot provided either positive emotional feedback, negative emotional feedback or neutral feedback. They found that the robot providing positive emotional feedback positively influenced children's learning gain and their social engagement with the robot. They did not investigate, however, children's engagement with the task and the question arises as to whether the same effects can be found with younger children, who are shown to rely more on the experimenter than the robot [44]. In a previous study [7], we investigated younger children's task engagement as well as robot engagement. In this study, we examined the effect of providing feedback on five-year-old children's task engagement, robot engagement and learning gain. Similar to Ahmad and colleagues, the

children played a game with a robot and a tablet. The children played with three robots, one providing feedback based on feedback approved by teachers, one providing feedback that aligns with what teachers would use and a robot providing no feedback. We found that the children were more task-engaged and robot-engaged if the robot provided feedback (in both feedback conditions) than with a robot that did not provide feedback. However, children did not learn more in the different conditions, nor did we explicitly test the effect of implicit or explicit feedback on children's engagement or learning gain. Both our previous study and the study by Ahmad et al. [9] used a tablet as an interaction medium between the robot and child. However, a disadvantage of using a tablet is that it can play a large role in the interaction [45,46] and reduce the children's attention to the robot tutor, which can lead to a decrease in children's robot engagement and their learning gain. It is, therefore, interesting to investigate the influence of robot feedback on children's task engagement and robot engagement during a robot tutoring lesson without a tablet present.

### 1.2. This Study

In this study, children received a tutoring lesson from a social robot and learned how to count in English using physical blocks. We investigated whether children were more task-engaged and more robot-engaged with a robot providing adult-like feedback (implicit negative feedback and explicit positive feedback), peer-like feedback (explicit negative feedback) or no feedback, and whether eye-gaze direction can predict task engagement or robot engagement. Finally, we investigated the relation between children's task engagement and robot engagement with children's learning gain. We addressed the following hypotheses:

**Hypothesis 1.**

(a)  *Children are more task-engaged with a robot that provides feedback than with a robot that does not provide feedback.*
   *We expect that children's task engagement will be higher when children receive feedback because the feedback will make them aware of their mistakes. This awareness can lead to a more successful completion of the task and children's success will result in higher task engagement.*
(b)  *Children are more robot-engaged when the robot provides adult-like feedback than in the other two conditions. We expect this result because the adult-like feedback is the only condition that provides positive feedback, which is shown to increase children's motivation and can increase children's robot engagement [47,48]. We expect that this effect will mainly contribute to children's robot engagement because the robot is providing the positive feedback and children might like the robot more due to these positive expressions.*

**Hypothesis 2.**

(a)  *Eye gaze toward the blocks and the robot has a positive relation with children's task engagement and children's eye gaze elsewhere has a negative relation with children's task engagement.*
   *We expect that this is because the task involves both the robot as an instructor and the blocks because the children have to manipulate these blocks during the task.*
(b)  *Children's eye gaze toward the robot will have a positive relation with robot engagement and the other eye-gaze directions will have a negative relation with robot engagement.*
   *We expect that only eye gaze toward the robot will have a positive relation with robot engagement, because when you communicate and, therefore, engage with a robot as a social partner, this is often accompanied by mutual eye gaze with this social partner [49] and other studies that detected disengagement with the robot [30–32] when participants looked away.*

## 2. Method

A between-subjects design with three conditions was employed for this study. Children received either adult-like feedback, peer-like feedback or no feedback. The robot behavior remained the same through the conditions except for the robot's feedback.

### 2.1. Participants

A total of 58 native Dutch children ($M_{age}$ = 3 years and 6 months, $SD_{age}$ = 4 months) participated in this study. All children attended a preschool or childcare in the Netherlands. For all children, the parents signed an informed consent form to give permission. The participants were randomly distributed over the three conditions. Four children indicated that they wanted to stop participating during the experiment and therefore stopped the experiment prematurely and were removed from the data. This resulted into the following distribution:

1.  Adult-like feedback ($N$ = 21, $M_{age}$ = 3 years and 6 months, 12 boys, 9 girls);
2.  Peer-like feedback ($N$ = 18, $M_{age}$ = 3 years and 6 months, 10 boys, 8 girls);
3.  No feedback ($N$ = 19, $M_{age}$ = 3 years and 7 months, 13 boys, 6 girls).

Exact age data for four children are missing and are not included in the age calculation. The study was conducted in accordance with the Declaration of Helsinki and received ethical approval from the Research Ethics committee of Tilburg School of Humanities and Digital Sciences.

### 2.2. Robot Tutoring Lesson

The interaction was completely in Dutch, except for the target words, which were in English (the target words are italicized in this section to indicate which words were spoken in English). The aim of the lesson was to teach children to count from one to four in English. Before the tutoring lesson, children participated in a group introduction and received a pre-test. The tutoring lesson started with the robot teaching the children the four counting words using different training tasks. These training tasks varied from repeating the target words, counting various body parts of the robot to building a tower with blocks and counting the height of the tower. For instance, the robot would ask the child to build a tower and to count together how tall the tower is: "Shall we count together in English how tall this tower is? Repeat after me: *one, two, three, four*." (in Dutch: "Zullen we samen tellen hoe hoog de toren is in het Engels? Zeg mij maar na: *one, two, three, four*."). All target words were repeated three times during these training tasks. After this concept binding of the target words, the robot and child went over the different target words with the use of the four blocks. For each target word, the robot asked the child to collect a certain number of blocks using an English counting word: "I'm going to say in English how many blocks you should grab: *three*" (in Dutch: "Ik ga in het Engels zeggen hoeveel blokken jij mag pakken: *three*"). The order of the target words was fixed and was, therefore, the same for each child. Each target word was asked only once during these practice rounds to reduce the duration of the experiment. Once the child collected the blocks, the robot provided feedback (only in the adult-like and peer-like feedback conditions) and continued with the next instruction. After all words were practiced, the robot and child concluded the session with a Dutch children's dance.

### 2.3. Experimental Conditions

The children received either adult-like feedback, peer-like feedback, or no feedback (see Table 1 for an example):

1.  In the adult-like feedback condition, the robot used explicit positive feedback for correct answers and implicit negative feedback for incorrect answers. A correct answer would invoke a facial expression using colored eye-LEDs and positive verbal feedback ("That is right, *three* means three in English"). For an incorrect answer, corrective feedback was provided ("*three* means three"). After receiving negative feedback, children could try again ("You should take *three* blocks"), after which the robot would again provide feedback. This negative feedback was, at most, provided twice for every target word, which means that during the experiment, every child was able to receive negative feedback eight times and positive feedback four times. In case the child gave more than two incorrect answers, the robot still provided

positive feedback and continued to the next instruction. For both positive feedback and negative feedback, the robot repeated the English target word, which increased children's exposure to the target words.

2.  In the peer-like feedback condition, the robot did not provide positive feedback but only provided explicit negative feedback. This explicit negative feedback was based on children's feedback during peer interaction [50]. Similar to the adult-like feedback condition, children could try again twice after receiving negative feedback. After a correct answer, the robot would continue to the next step without any feedback.

3.  In the no feedback condition, the robot did not provide any feedback and just continued the game with the blocks after children collected the correct or incorrect number of blocks.

**Table 1.** An example of the robot's feedback in the different feedback conditions.

| | Correct Answer | | Incorrect Answer | |
|---|---|---|---|---|
| **Condition** | **Dutch** | **English** | **Dutch** | **English** |
| Adult-like | Dat is goed! Three betekent drie in het Engels. | That is right! *Three* means three in English | Three betekent drie, je moet drie blokken pakken. Probeer opnieuw | *Three* means three, you should take three blocks. Try again |
| Peer-like | - | - | Dat is fout! Je moet drie blokken pakken. Probeer opnieuw. | That is wrong! You should take three blocks. Try again. |
| No feedback | - | - | - | - |

### 2.4. Materials

#### 2.4.1. Experimental Setting

The experiment took place in multiple preschools and childcare centers in the Netherlands. At each location, the experiment room was a classroom that the children were familiar with, but not in use by the school. The Softbank Robotics NAO robot was used, which is commonly deployed in experiments with children. Moreover, four blue blocks were used. We chose to use blocks in our experiment because preschool children are used to playing with blocks, and children learn how to manipulate and handle blocks to enhance their visual-spatial skills [51]. The children sat on the ground with the crouched robot, approximately 40 cm from each other, with the blocks in between (see Figure 1 for the experimental setting). The children were positioned so they could not see the corridor and, therefore, could not see other children passing by the room. The children were filmed from two viewpoints: one camera was positioned in front of the child to record his or her face and one camera was sideways to record the social interaction between robot and child. Two experimenters were present during the interaction to operate the robot and to provide reassurance for the children if necessary. While the experimenters sometimes instructed the child to perform a task if required, they were careful not to provide feedback.

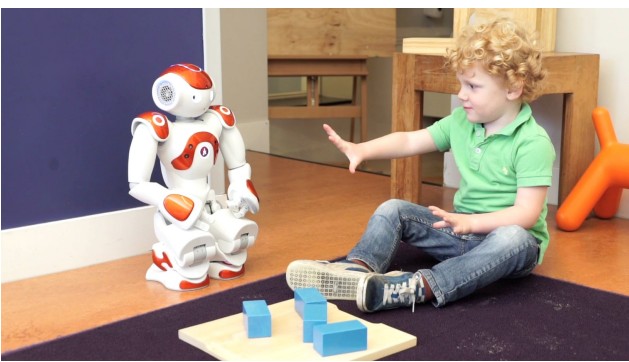

**Figure 1.** The setup of the experiment.

### 2.4.2. Pre-Test

Before the child started the tutoring sessions, his or her Dutch and English knowledge of the four target words were tested. The experiment leader asked the child to collect a number of blocks and repeated this for every target word (e.g., "Can you give me four blocks?"). This process was first completed in Dutch to test their L1 knowledge of the target words, and then in English to test their L2 knowledge. The same blocks were used as during the tutoring session; however in this case six blocks were used instead of four to reduce the chance of guessing. For both Dutch and English, the experimenter noted how many target words the child already knew in both languages. The experimenter did not provide feedback between the words, and only continued with the next target word.

### 2.4.3. Post-Test

The post-test was the same as the pre-test; however it was only conducted for the English target words. The experimenter used six blocks and asked the children to collect the number of blocks that was equal to each of the four target words.

### *2.5. Procedure*

### 2.5.1. Group Introduction

The study consisted of two group introductions and one tutoring session. One week before and in the morning of the experimental day, group introductions were given to familiarize the children with the robot and build up trust and rapport with the robot [52]. All children in the classroom participated during the first group introduction, but only the children that participated in our experiment joined the second introduction. Both introductions were the same, and during these we highlighted some of the similarities of the robot with people to establish common ground, since this can have a positive effect on the learning outcome [24]. For example, we explained that the robot has arms and legs just like people have and can express emotions through its eye-LEDs. The robot and children would then dance a familiar Dutch children's song. We never forced the children to participate; if they declined they could sit in a quiet corner and watch from a distance.

### 2.5.2. Experiment

After the children were brought to the experiment room, the experimenter tested their prior knowledge of the target words in both Dutch and English, as described in Section 2.4.2. The pre-test was carried out in the same room as the tutoring session, but at some distance from the robot. After the pre-test, the child was asked to sit in front to the robot. During the experiment, the two experimenters remained in the room at a distance to discourage children from looking at them. When children looked at the experimenters or asked them for help, the experimenters redirected the children's attention back to the robot. When a child displayed signs of discomfort, the experimenters comforted the child and tried to make him or her more relaxed. For some children, the experimenters remained close to the children and helped them during the beginning of the interaction. In the case of four children, the experiment was stopped and these children were brought back to their classrooms.

After the robot tutoring lesson was completed, the experiment finished with an English post-test. When this post-test was completed, a short debriefing was conducted. During this debriefing, the experimenters repeated all of the target words and their translations to ensure that children had learned the correct translation. Finally, the child was brought back to their classroom. The duration of the experiment was approximately 10 to 15 min.

### *2.6. Engagement and Gaze Coding*

We manually coded three different aspects of the interaction: task engagement, robot engagement and children's eye-gaze direction. Not all of interaction was coded; instead, we chose two video fragments: one two-minute fragment at the beginning of the interaction, and a two-minute fragment at the end of the interaction. The gaze coding was only

completed for the two-minute fragment at the end of the interaction due to time constraints. The two video fragments of the interaction were chosen to code different aspects in the interaction, with the first fragment being the moment when the robot started to teach the children English, and the other fragment being a moment in the end of the interaction when the robot and child started to play with the blocks. These fragments resulted in 116 video fragments for 58 children.

### 2.6.1. Engagement Coding

Both task engagement as well as robot engagement were rated on a Likert scale from one to five, including half points, with one being a low level of engagement and five highly engaged. We based our engagement coding scheme on an existing coding scheme named ZIKO [53]. This coding scheme is used in children's day cares to measure, among other things, children's engagement to improve the day care activities. We adapted the scheme to include specific cues for our own experiment, such as attention toward the experiment leader instead of the robot and blocks. Children were fully task-engaged when they were completely "absorbed" in the robot-block activity, when they showed to be open for new information, were very motivated and listened to the tasks. Robot engagement described children's engagement with the robot as a social partner and focused more on the interaction itself than on the task. Each engagement level had specific cues for the rater to look for.

A *high task engagement* had cues such as: looking at the task and robot, actively answering and grabbing blocks, listening for new instructions and being fully committed to the task. In contrast, a *low task engagement* was indicated by fiddling, not performing, and playing with objects not related to the task (e.g., their shoes). A *neutral task engagement* was determined as the child executing the tasks but not being fully immersed in them.

A *high robot engagement* had cues such as: looking at the robot, having an open body posture toward the robot, having spontaneous conversations with the robot. A *low robot engagement* had cues such as: turning away from the robot. A *neutral robot engagement* had cues such as touching the robot without meaning. For all specific cues and information, see the coding scheme on Github https://www.github.com/l2tor/codingscheme (accessed on 1 June 2021).

Ten percent of the data were coded by two raters and their inter-rater agreement was considered moderate using the intraclass correlation coefficient ($ICC_{task} = 0.75$, 95% CI—[0.05, 93], $ICC_{robot} = 0.64$, 95% CI—[0.16, 0.88]) [54].

### 2.6.2. Eye-Gaze Coding

We coded children's eye gaze toward different directions in order to measure their visual attention using ELAN [55]. We analyzed the same fragments as engagement, but only the second fragment when children also used the blocks for the interaction. In particular, we coded children's eye gaze in five different directions: towards the robot, blocks, experimenter, elsewhere and unknown. The latter direction unknown (0.71%) was not included in the analysis. Eye gazes that were shorter than one second were excluded and added to the nearest annotation, as a short glance would not change the children's focus point. For the analyses, we calculated the duration for each category. To assess inter-rater reliability for this categorical data, 10% of the videos were coded by a second annotator, yielding a Fleiss' Kappa of 0.83, which is considered a very good agreement.

### 2.7. Analyses

We investigated children's task engagement and robot engagement over the lesson and the conditions. We measured the two engagement types in the beginning of the lesson and the end of the lesson.

To inspect the normality of the engagement data, Q-Q plots were plotted and the Shapiro–Wilk test was conducted. Both the plots and Shapiro–Wilk tests showed a non-normal distribution of the task engagement and robot engagement. Consequently [56], we

conducted two robust two-way mixed design ANOVAs with 20% trimmed means and the feedback condition as a between-subjects variable and the test moment (beginning and end of lesson) as a within-subjects variable on both engagement types. We used the "WRS2" R package to conduct this analysis [57].

To investigate the relation between children's eye-gaze direction and their engagement, multiple regression analyses of task engagement and robot engagement were performed using four predictors: duration of eye-gaze toward the blocks, the robot, the experimenter and elsewhere. The assumptions of non-multicollinearity were checked using variance inflation factor (VIF) statistics [58]. Several models were analyzed, from which the best model was chosen.

We investigated the effect of the different feedback types on children's learning gain. A Q-Q plot and a Shapiro–Wilk tests showed a non-normal distribution of the learning gain. Therefore, we conducted a robust mixed design ANOVA with 20% trimmed means to test the effect of the tutoring lesson and feedback on children's word knowledge scores.

Finally, we investigated the relation between engagement and learning using a Pearson correlation analysis.

## 3. Results

First, we will report on the effects of the experimental conditions on children's task engagement and robot engagement. Next, we will discuss the relation between children's eye-gaze direction and their engagement. Finally, we will report the effect of the three feedback conditions on children's learning gain and the relation of learning gain and engagement.

### 3.1. Engagement

To begin, we investigated whether task engagement and robot engagement were related. Task engagement and robot engagement were correlated ($r(218) = 0.70, p < 0.001$), indicating that children who scored higher on task engagement also scored higher on robot engagement.

### 3.1.1. Task Engagement

We investigated whether the three experimental feedback conditions had an effect on children's task engagement. We expected that children would be more task-engaged when the robot was providing feedback. Figure 2a shows that there were large individual differences in children's task engagement over time, conditions and between the individual children. Some children became more task-engaged over time (48%), other children became less task-engaged over time (38%) and other children were equally engaged in the beginning of the lesson as in the end (14%). When looking at the graph, on average children scored higher than a neutral task engagement (3.0), except at the beginning of the lesson for the peer-like feedback condition.

We carried out a robust two-way mixed design ANOVA using trimmed means on children's task engagement with condition as between factor and test moment (beginning of the lesson and end of the lesson) as within factor. In contrast to our expectations, there was no significant difference between children in the different conditions ($F(2, 22.67) = 0.58, p = 0.57$), nor was there a significant difference over time ($F(1, 33.37) = 0.12, p = 0.73$). However, there was a significant interaction effect between condition and test moment ($F(2, 23.17) = 6.89, p < 0.01$). This interaction effect is illustrated in Figure 2a; children's task engagement in the peer-like and in the adult-like feedback conditions increased during the lesson and in the no feedback condition their task engagement decreased over time.

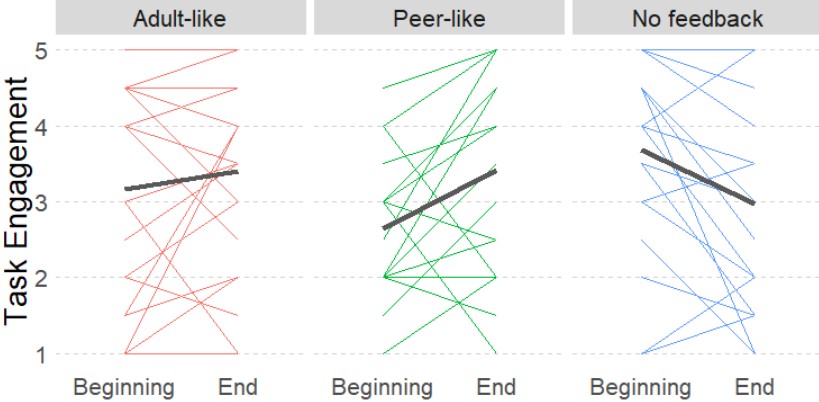

(**a**)

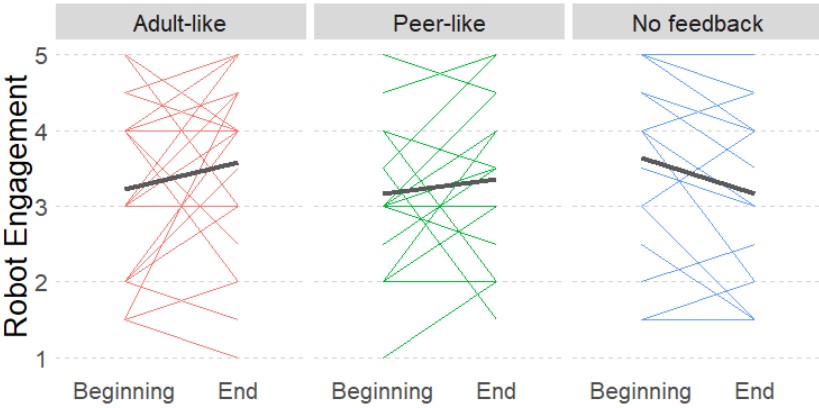

(**b**)

**Figure 2.** The individual children's (**a**) task engagement scores and (**b**) robot engagement scores over the three conditions in the beginning and end of the tutoring lesson. The dark lines show the averages of the children's engagement scores.

### 3.1.2. Robot Engagement

Similar to children's task engagement, children's robot engagement varied for each condition in the beginning and end of the lesson (see Figure 2b). It decreased for 45% of the children, increased for 43% of the children and remained the same for 12% of the children. To investigate whether their robot engagement was different for the three feedback conditions, we conducted a robust two-way mixed design ANOVA using trimmed means on children's robot engagement with condition as between factor and test moment (beginning of the lesson and end of the lesson) as within factor. We expected that children would be more robot-engaged when interacting with a robot providing adult-like feedback than the other two conditions. Contrary to these expectations, there was no significant difference between children in the different conditions ($F(2, 22.57) = 0.16$, $p = 0.85$), nor a difference over time ($F(1, 31.57) = 0.01$, $p = 0.93$). Similar to task engagement, there was a significant interaction effect between condition and test moment ($F(2, 22.36) = 3.88$, $p = 0.04$). This means that children's robot engagement was influenced by the three feedback conditions and the moment in the lesson. Figure 2 shows that children's robot engagement increased during the two feedback conditions and decreased in the no feedback condition. This increase in robot engagement appeared to be less strong than with task engagement.

### 3.2. Duration of Eye-Gaze Directions as Engagement Predictor

Next, we investigated whether the duration of children's different eye-gaze directions had a relation with children's engagement. Table 2 shows the duration in seconds toward the robot, human experimenter, the blocks and elsewhere in the different conditions. Overall, children spent the most time looking at the blocks, followed by the robot, the experimenter and they spent the least time looking elsewhere. To investigate the relation between the duration of each eye-gaze direction and engagement, we carried out a linear regression to predict the role of eye gaze on children's task engagement and robot engagement.

**Table 2.** The mean duration in seconds for the children's eye-gaze direction divided into each feedback condition (SD between brackets).

| Condition | Robot | Blocks | Experimenter | Elsewhere |
|---|---|---|---|---|
| Adult-like | 45.1 (21.0) | 57.1 (15.3) | 13.5 (12.0) | 2.7 (3.4) |
| Peer-like | 38.1 (24.2) | 57.2 (30.1) | 15.4 (15.2) | 2.4 (3.3) |
| No feedback | 31.6 (19.3) | 56.0 (18.5) | 18.6 (17.5) | 5.6 (7.9) |
| Overall | 38.1 (21.9) | 56.8 (21.6) | 15.9 (15.0) | 3.6 (5.5) |

### 3.2.1. Task Engagement

Table 3 shows the different regression analyses we performed. Model 1 included all eye-gaze directions and when combined, these explained a significant proportion of the variance of task engagement ($F(4, 50) = 16.13, p < 0.001, R^2_{adj} = 0.53$). However, when checking for multicollinearity, we found that the duration that children looked toward the blocks and toward the robot were highly related (VIF scores: *blocks* = 6.45, *robot* = 6.48) and strongly correlated ($r = -0.69, p < 0.001$). Following [58], we combined these two directions by taking the sum of the two directions and using their total duration (*blocks and robot*) in a new model. Model 2 also explained a large proportion of variation ($F(3, 51) = 21.54, p < 0.001, R^2_{adj} = 0.53$) with acceptable VIF values (VIF scores: *blocks and robot* = 3.86, *experimenter* = 3.22, *elsewhere* = 1.54). As an alternative to using the total duration in both eye-gaze directions, Model 3, we removed the predictor with the highest VIF value [58], which was the duration children looked at the robot. In this alternative model, the duration that children looked at the blocks did not contribute significantly to the prediction. Hence, we removed this variable from the model. The resulting Model 4 significantly explained 48% of the task engagement's variance and did not perform better than the other models.

Therefore, the best model was Model 2 ($R^2_{adj} = 0.53$) and the resulting regression equation was:

$$Eng_{task} = 8.89 - 0.04 \times Gaze_{blocks\ and\ robot} - 0.09 \times Gaze_{experimenter} - 0.10 \times Gaze_{elsewhere} \tag{1}$$

where $Eng_{task}$ is task engagement, $Gaze_{blocks\ and\ robot}$ the duration in seconds of eye-gaze toward the blocks and the robot, $Gaze_{experimenter}$ is the duration toward the experimenter and $Gaze_{elsewhere}$ is the duration that children looked elsewhere.

**Table 3.** Regression analyses summary for the duration (s) that children looked in different directions predicting children's task engagement.

| Eye-Gaze Direction | Coefficient | SE | VIF | t | p |
|---|---|---|---|---|---|
| **Model 1** | | | | | |
| *constant* | 8.93 | 1.66 | | 5.39 | <0.001 |
| robot | −0.04 | 0.02 | 6.48 | −2.55 | 0.01 |
| blocks | −0.04 | 0.02 | 6.45 | −2.86 | 0.01 |
| experimenter | −0.09 | 0.02 | 3.23 | −5.75 | <0.001 |
| elsewhere | −0.10 | 0.02 | 1.55 | −4.84 | <0.001 |
| **Model 2** | | | | | |
| *constant* | 8.89 | 1.65 | | 5.40 | <0.001 |
| blocks and robot | −0.04 | 0.01 | 3.86 | −2.78 | 0.01 |
| experimenter | −0.09 | 0.02 | 3.22 | −5.80 | <0.001 |
| elsewhere | −0.10 | 0.02 | 1.54 | −4.83 | <0.001 |
| **Model 3** | | | | | |
| *constant* | 4.87 | 0.48 | | 10.20 | <0.001 |
| blocks | −0.01 | 0.01 | 1.14 | −1.23 | 0.22 |
| experimenter | −0.06 | 0.01 | 1.07 | −6.03 | <0.001 |
| elsewhere | −0.07 | 0.02 | 1.06 | −3.92 | <0.001 |
| **Model 4** | | | | | |
| *constant* | 4.34 | 0.21 | | 20.80 | <0.001 |
| experimenter | −0.06 | 0.01 | 1.00 | −5.88 | <0.001 |
| elsewhere | −0.07 | 0.02 | 1.00 | −3.71 | <0.001 |

Model 1: $Eng_{task} = \alpha + \beta \times \text{robot} + \beta \times \text{blocks} + \beta \times \text{experimenter} + \beta \times \text{elsewhere}$, $F(4, 50) = 16.13$, $p < 0.001$, $R^2_{adj} = 0.53$, $RSE = 0.29$, Model 2: $Eng_{task} = \alpha + \beta \times \text{blocks and robot} + \beta \times \text{experimenter} + \beta \times \text{elsewhere}$, $F(3, 50) = 21.54$, $p < 0.001$, $R^2_{adj} = 0.53$, $RSE = 0.29$, Model 3: $Eng_{task} = \alpha + \beta \times \text{blocks} + \beta \times \text{experimenter} + \beta \times \text{elsewhere}$, $F(3, 51) = 17.45$, $p < 0.001$, $R^2_{adj} = 0.48$, $RSE = 0.30$, Model 4: $Eng_{task} = \alpha + \beta \times \text{experimenter} + \beta \times \text{elsewhere}$, $F(2, 52) = 25.18$, $p < 0.001$, $R^2_{adj} = 0.47$, $RSE = 0.30$.

### 3.2.2. Robot Engagement

For robot engagement we used a similar approach as for task engagement. We performed different multiple regression models to predict children's robot engagement using the duration of children's eye gaze toward the blocks, the robot, the experimenter and elsewhere. Similarly to task engagement, the model containing all variables explained a significant proportion of the variance of children's robot engagement (see Table 4 for the models). Model 1 showed that 58% of the variance in children's robot engagement can be explained by the duration in which children looked at the four different eye-gaze directions. However, both the duration that children looked in the direction of the robot and in the direction of the blocks did not significantly contribute to the model and could therefore be removed. We ran three further models: Model 2 without children's eye-gaze direction toward the robot, Model 3 without children's eye gaze toward the blocks and Model 4 without both the eye gaze toward the robot and blocks. Model 4 contained the fewest predictors, but also explained the lowest variance of the four models ($R^2 = 0.45$). Despite this, the other two models (3 and 4) were the same regarding the variance ($R^2 = 0.58$), we prefer the model containing gaze toward to the robot instead of blocks because this model (Model 3) shows the positive relation between eye-gaze direction to the robot and robot engagement.

The resulting regression equation for robot engagement was:

$$Eng_{robot} = 3.35 + 0.02 \times Gaze_{robot} - 0.04 \times Gaze_{experimenter} - 0.03 \times Gaze_{elsewhere} \tag{2}$$

where $Eng_{robot}$ is robot engagement, $Gaze_{robot}$ the duration in seconds that children looked at the blocks, $Gaze_{experimenter}$ is the duration that children looked at the experimenter and $Gaze_{elsewhere}$ is the duration of children's eye gaze elsewhere.

**Table 4.** Regression analyses summary for the duration (s) that children looked in different directions predicting children's robot engagement.

| Eye-Gaze Direction | Coefficient | *SE* | *VIF* | *t* | *p* |
|---|---|---|---|---|---|
| **Model 1** | | | | | |
| *constant* | 4.87 | 1.31 | | 3.71 | <0.001 |
| robot | 0.01 | 0.01 | 6.48 | 0.61 | 0.55 |
| blocks | −0.01 | 0.01 | 6.45 | −1.18 | 0.24 |
| experimenter | −0.05 | 0.01 | 3.23 | −4.13 | <0.001 |
| elsewhere | −0.04 | 0.02 | 1.55 | −2.61 | 0.01 |
| **Model 2** | | | | | |
| *constant* | 5.63 | 0.36 | | 15.69 | <0.001 |
| blocks | −0.02 | 0.00 | 1.14 | −4.15 | <0.001 |
| experimenter | −0.06 | 0.01 | 1.07 | −8.10 | <0.001 |
| elsewhere | −0.05 | 0.01 | 1.06 | −3.59 | <0.001 |
| **Model 3** | | | | | |
| *constant* | 3.35 | 0.28 | | 11.81 | <0.001 |
| robot | 0.02 | 0.01 | 1.14 | 3.98 | <0.001 |
| experimenter | −0.04 | 0.01 | 1.14 | −5.33 | <0.001 |
| elsewhere | −0.03 | 0.01 | 1.01 | −2.36 | <0.001 |
| **Model 4** | | | | | |
| *constant* | 4.29 | 0.18 | | 24.09 | <0.001 |
| experimenter | −0.05 | 0.01 | 1.00 | −6.32 | <0.001 |
| elsewhere | −0.04 | 0.02 | 1.00 | −2.34 | 0.02 |

Model 1: $Eng_{robot} = \alpha + \beta \times \text{robot} + \beta \times \text{blocks} + \beta \times \text{experimenter} + \beta \times \text{elsewhere}$, $F(4, 50) = 19.48$, $p < 0.001$, $R^2_{adj} = 0.58$, $RSE = 0.22$, Model 2: $Eng_{robot} = \alpha + \beta \times \text{blocks} + \beta \times \text{experimenter} + \beta \times \text{elsewhere}$, $F(3, 51) = 26.17$, $p < 0.001$, $R^2_{adj} = 0.58$, $RSE = 0.22$, Model 3: $Eng_{robot} = \alpha + \beta \times \text{robot} + \beta \times \text{experimenter} + \beta \times \text{elsewhere}$, $F(3, 51) = 25.31$, $p < 0.001$, $R^2_{adj} = 0.58$, $RSE = 0.22$, Model 4: $Eng_{robot} = \alpha + \beta \times \text{experimenter} + \beta \times \text{elsewhere}$, $F(2, 52) = 23.35$, $p < 0.001$, $R^2_{adj} = 0.45$, $RSE = 0.25$.

### 3.3. Learning Gain

Next, we examined whether different forms of feedback influenced children's word knowledge. Table 5 reveals that children on average knew between one and two words after the lesson, but standard deviations are high. Children performed above chance level in the pre-test (chance level = 0.16, $W = 4879$, $p < 0.001$) and post-test (chance level = 0.16, $W = 4945$, $p < 0.001$). A robust mixed-design ANOVA with 20% trimmed means and with children's word knowledge as dependent variable and with condition as between factor and the two test moments (pre- and post-test) as within variable showed that children did not know significantly more words ($M = 1.43$, $SD = 0.95$) after the lesson than before the lesson ($M = 0.98$, $SD = 0.65$; $F(1, 17.71) = 3.76$, $p = 0.07$). There were no significant differences between the three conditions ($F(2, 14.94) = 1.81$, $p = 0.20$) nor a significant interaction effect between conditions and test moment ($F(2, 14.94) = 0.05$, $p = 0.95$). This showed that children did not know significantly more target words after the lesson than before the lesson, independent of the condition.

**Table 5.** The children's average word knowledge scores on the pre-test and post-test for the three conditions (SD between brackets).

| Feedback | Pre | Post |
|---|---|---|
| Peer-like | 1.18 (0.7) | 1.61 (0.9) |
| Adult-like | 0.90 (0.4) | 1.38 (1.0) |
| No | 0.91 (0.8) | 1.33 (0.9) |
| Total | 0.98 (0.7) | 1.43 (0.9) |

### 3.4. Relation Learning Gain, Task Engagement and Robot Engagement

Finally, to investigate whether there is a relation between L2 word knowledge and children's task engagement and robot engagement, we performed a Pearson correlation

analysis. We did not find any significant correlation between children's learning gain and task engagement ($r(109) = 0.12$, $p = 0.21$). Likewise, we did not find a significant correlation between robot engagement and learning gain ($r(109) = 0.16$, $p = 0.10$), meaning that children's engagement levels did not have a relation with how many words children learn.

## 4. Discussion

In this paper, we present a study in which we investigated the role of robot feedback on toddlers' task engagement and robot engagement, their learning gain and the relation between toddlers' eye-gaze direction and engagement. The children were assigned to one of three feedback conditions: a robot providing feedback like an adult would (adult-like feedback), a robot providing feedback like a peer would (peer-like feedback) and a condition where the robot provided no feedback (no feedback). While task engagement and robot engagement are different concepts, they are moderately correlated and show similar trends. Both engagement types decreased when children did not receive any feedback and increased during the lesson for peer-like feedback and adult-like feedback. Moreover, for both engagement types there were large individual differences between children. Given these similar trends for the two engagement types, we will first discuss the results combined and then discuss the differences in our findings between these two engagement types.

### 4.1. Engagement

We investigated children's task engagement and robot engagement in the beginning and the end of the tutoring session with the robot. Overall, children were engaged with the task and robot, and their engagement remained approximately the same over time. Contrary to our expectations, there was no main effect of feedback on children's *task engagement* (H1a) nor on children's *robot engagement* (H1b). We did not expect that because we did find an effect of feedback on children's task engagement and robot engagement in our previous study [7]. This difference may be explained by the fact that, in the current study, the robot's behavior did not differ sufficiently in the three conditions compared to our previous study. The current study only provided a limited number of exposures to the target words while in our other study the robot repeated each target word ten times. Hence, there might not have been enough feedback moments in order to observe a significant effect across the different conditions. Although this is a limitation of our design, we did not want to increase the duration of the session because children's attention span at this young age is very short [59]. In future investigations, it might be recommended to use multiple sessions with these young children, to measure an effect on children's task engagement and robot engagement. There are, however, other possible explanations. The result might also be explained by the fact that the children in our current study were younger than in our previous experiment: we investigated 3–4-year-old children rather than 5–6-year-old children in our previous study. Children undergo major developmental progress at this age and learn how to think more logically when they get older [60]. It is possible that younger children need more, or other types of feedback than older children. Another possible explanation for our findings is that the individual differences between children are larger than the differences between the conditions. As Figure 2 shows, there were many individual differences between children, which is in line with many other studies [7,61]. It is possible that some children would have been more engaged with a robot providing peer-like feedback and other children with adult-like feedback. Our study did not include enough participants to investigate these individual differences, and future studies with more participants will need to be undertaken.

Furthermore, there was no main effect of time on task engagement and robot engagement, which is, again, surprising because in previous experiments children's task engagement dropped over time within one lesson [10,62]. It is possible that this is due to the duration of the lesson: our lesson was shorter than those of de Wit et al. [10] and van Minkelen et al. [62] due to the shorter attention span of the children, which might explain

the difference between the previous studies and the current one. It is also possible that there was too much variation between children and conditions, that nullified the effects over time. Finally, a specific explanation for the lack of results for *task engagement*, is that the beginning of our task itself (counting together with the robot) was very different than the end (playing with the blocks) and that this variation kept children task-engaged. In the two experiments by de Wit et al. [10,28], the task remained the same during the full session and it is likely that children's task engagement dropped due to the lack of variation [63]. Our expectation is that this game variation will mostly influence task engagement; however, since we did not investigate this, it is possible that it will also influence robot engagement, e.g., because the robot's instructions are more important during one aspect of the task and as a result children look more at the robot which will increase their robot engagement.

While we did not find a main effect of conditions or time on engagement, we did find an interaction effect of condition and time. When inspecting Figure 2, we can observe that children's task engagement increased for both feedback conditions and it decreased in the no feedback condition over time. We saw a similar pattern for robot engagement. It is likely that children in the no feedback condition became less task-engaged and robot-engaged during the lesson because this condition did not include any feedback whether they completed the tasks successfully or unsuccessfully. The absence of positive confirmation when children accomplished the task may have played a role in their task engagement and robot engagement and therefore may have reduced it [47,48]. In a similar way to how the absence of corrective feedback to help the children in the rest of the session might have reduced their attention for the learning task, it also possibly increased frustration [64], which consequently could have led to task disengagement. Thus, feedback seems to have a positive effect on children's engagement over time.

*4.2. Duration of Eye-Gaze Directions as Engagement Predictor*

We explored the relation between eye gaze and children's task engagement and robot engagement in order to understand whether this important but single aspect of engagement can successfully predict task engagement and robot engagement.

Our findings showed that children's eye-gaze direction can explain a large proportion of the variance of both children's task engagement and robot engagement. In particular, children's task engagement had a negative relation with the duration children looked at the robot and blocks combined, and with the duration children looked at the experimenter and elsewhere. There were multiple models possible for robot engagement: (1) a negative relation with the duration children looked at the *blocks, the experimenter* and *elsewhere* and (2) a positive relation with the duration children looked at the *robot*, and a negative relation with the duration children looked at the *experimenter* and *elsewhere*. These results might seem surprising; however, when looking at the regression equations they can be explained. For children's *task engagement*, all gaze directions were taken into account in the equation. Our expectation (H2a) was that the duration that the children looked at the blocks and at the robot would have a positive relation with children's task engagement and the duration that children looked at the experimenter or elsewhere a negative relation. Our regression equation showed that the duration that children looked at the experimenter and elsewhere would lower the rate of task engagement more (with factors of 0.09 and 0.10, respectively) than the duration that children looked at the blocks and robot combined (0.04). The larger role of looking elsewhere (and perhaps looking at the experimenter) supports previous studies that used eye gaze to detect disengagement and as a cue to initiate different robot behaviors that can re-grab the participant's attention [31]. It is possible that children's disengagement (attention away from the task and directed at the experimenter + elsewhere) is easier to detect using eye gaze and can be used in future studies to initiate engagement-increasing behaviors in the robot.

Moreover, for children's *robot engagement*, there were two models performing equally well: a model including the duration that children looked at the blocks, experiment and elsewhere and a model including the duration that children looked at the robot,

experiment and elsewhere. The model including robot gaze had a positive relation with robot engagement (H2b), and the model including eye gaze toward to blocks had a negative relation with robot engagement. These can both be explained by examining the regression coefficients. When gaze toward the robot is not included, all other eye-gaze directions have a negative effect on robot engagement. However, when robot gaze is included, this eye-gaze direction has a positive relationship with robot engagement. Therefore, even though both models will explain robot engagement equally well, we prefer to use the model containing eye gaze toward the robot because it follows intuitively that eye gaze at the robot predicts robot engagement.

Our results provide further support for the hypothesis that eye gaze is a good predictor for task engagement and robot engagement and that future studies can use eye gaze for automatic systems to detect engagement. These studies might additionally incorporate the robot's on-board camera to measure children's gazes in order to reduce the extra hardware needed. However, this can be complicated because the robot's head often moves.

Eye gaze explained a larger proportion of the variance for robot engagement than task engagement, a possible explanation is that robot engagement is a social engagement, and social interaction is often based on eye gaze toward each other [49]. A note of caution is due here since not all of the variance can be explained by eye gaze (task engagement (53%) and for robot engagement (58%)) which indicates that eye gaze does not predict every aspect of children's engagement (both task and robot). For task engagement elements such as children's speech, emotional expressions, fiddling or children's interaction with the blocks should be included and for robot engagement elements such as speech toward the robot, smiles during the conversation and body posture can be considered as predictors.

### 4.3. Learning Gain

Contrary to our expectations, we found that children did not learn during the interaction nor was this dependent on the condition. Children knew, irrespective of the condition, no more target words after the experiment than before the experiment. It is likely that the exposure to each target word was not enough, which reduced the training of target words and therefore children's learning gain. Three- and four-year-old children have a limited attention span of 3 to 4 min [65] and although our experiment lasted already much longer, we did not want to exhaust the children by introducing more repetitions. To create a more successful tutoring session, the target word exposure should be higher. Moreover, it should be noted that the children's general word knowledge after the sessions was low for all three conditions, which is a result more commonly found after robotic tutoring sessions [2]. Future studies should look at repeating target words over sessions, and perhaps focus on the words children did not know yet in the earlier sessions instead of repeating all words (creating a more personalized interaction). However, more exposure to target words is not necessarily the only answer, as indicated by the results of de Haas et al. [7] who investigated a robot providing feedback to five- to six-year-old children over three sessions. These children should have a longer attention span and target words were repeated 10 times during the sessions and they still did not find a learning difference between conditions. However, they did find a learning effect over the lessons.

There were large individual differences between children: some children learned all the target words and some did not learn any words. These individual differences is in line with previous research [66], where we specifically investigated the individual differences between preschoolers learning with a robot and found that the robot gestures benefited children's word knowledge in different ways across children. In our current study, it is possible that some children benefited from the adult-like feedback, and others from peer-like feedback.

### 4.4. Individual Differences

As already mentioned, there were large individual differences between children. Interestingly, when looking at children's responses in an exploratory manner, there is an

overlapping pattern for each condition. For example in the peer-like feedback condition, although the robot instructed the children to collect a certain number of blocks, a third of the children misunderstood the robot and simply repeated the target word (seven children) or repeated the word while also collecting the blocks (five children). This observation may be explained by the fact that the child had to repeat the word to the robot during the word concept binding phase of the interaction and they got used to repeating the L2 word when the robot used this L2 word. A similar variation was observed in the adult-like feedback condition, instead of collecting the blocks after the robot's instructions, three children built a tower, six repeated the robot and the experimenter had to intervene five times.

Moreover, children frequently requested additional support from the experimenter after the robot's instruction. Some children hardly looked at the robot and always looked at the experimenter while grabbing the blocks or needed additional persuasion to show the blocks to the robot. The experimenter intervened approximately four times during the whole tutoring lesson after the robot's instructions, this varied from repeating the robot's instruction, asking the children to grab the blocks instead of repeating the words, and instructing the children to pay attention to the robot.

Finally, some children started playing with the blocks and completely ignored the robot. For instance, they started to throw the blocks, to play with their shoes, and even started to play with the microphone close to the robot. The experimenters intervened when this happened and tried to redirect the child's attention to the robot, but some children lost their engagement completely. Occasionally, these children regained focus after the next instruction. This is probably due to the low attention span of this age group. This behavior is unfortunately inevitable, as there will always be children who have little attention for the task. Whether this is due to external factors, such as being fatigued or to the task itself is something that researchers should take into account when designing child–robot interactions.

Taken together, the children's responses after the robot instructions varied considerably. Other studies should, therefore, focus on personalizing the interactions for every child, even with preschool children like in our experiment [43,67]. In our experiment, we did not personalize the interaction in order to maximize the systematic effect of different feedback types on the children's task and robot engagement, although we did not find any differences.

## 5. Conclusions

Given the potential of social robots for tutors with preschool children, it is important to understand how children can be effectively tutored, while still being engaged with the task and robot. In this study, we investigated the effect of the robot's feedback on young children's task engagement and robot engagement in a second-language tutoring session. The robot either provided feedback as an adult, as a peer or no feedback during the tutoring session. Moreover, we explored the relation between eye-gaze direction and robot engagement and task engagement. Our findings showed that there was an interaction effect between children's engagement and the three feedback conditions. Providing feedback (as a peer and adult) increased children's task engagement and robot engagement during the lesson, while providing no feedback did not increase the task engagement and robot engagement. Finally, our study shows that children's eye-gaze direction is informative for children's task and robot engagement, which can contribute to automatic engagement measuring systems in child–robot tutoring interactions.

**Author Contributions:** Conceptualization: M.d.H., P.V. and E.K.; methodology: M.d.H.; software: M.d.H.; validation: M.d.H., P.V. and E.K.; formal analysis: M.d.H.; investigation: M.d.H.; resources: M.d.H.; data curation: M.d.H.; writing—original draft preparation: M.d.H.; writing—review and editing: P.V. and E.K.; visualization: M.d.H.; supervision: P.V. and E.K.; project administration: P.V. and E.K.; funding acquisition: P.V. and E.K. All authors have read and agreed to the published version of the manuscript.

**Funding:** This work was funded by the EU H2020 L2TOR project (grant 688014).

**Institutional Review Board Statement:** The study was conducted according to the guidelines of the Declaration of Helsinki, and approved by the Research Ethics Committee of Tilburg University (dated 15 January 2017).

**Informed Consent Statement:** Informed consent was obtained from all subjects involved in the study and their legal guardians.

**Data Availability Statement:** The data that support the findings of this study are available from the corresponding author upon reasonable request.

**Acknowledgments:** We thank Kinderopvanggroep Tilburg and all preschools for participating in this research. We are also grateful to Chiara de Jong and Peta Baxter for helping with the data collection.

**Conflicts of Interest:** The authors declare no conflict of interest.

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
