# Peer review of "When Preschoolers Interact with an Educational Robot, Does Robot Feedback Influence Engagement?"

_mti, doi:10.3390/mti5120077_

Round 1

Reviewer 1 Report

The article studies the effect of robot-children engagement on children learning experience. The children are from 3 to 4 years old, they are learning a second language (English). The study focus is on the role of feedback for guiding the learning process. The role of eye gaze engagement between the robot and children is also observed.

In general, the article is well written and easy to read, the authors might consider a proofread before submitting the final manuscript.

Regarding the methodology, the reviewer has the following concerns:

Regarding the” Experimental setting”, the authors used two cameras to film the children “The children were filmed from two viewpoints, one camera was positioned in front of the child to record his or her face and one camera was sideways to record the social interaction between robot and child”, both cameras are external to the robot, the question is: why the authors did not also use the camera integrated in the robot head? As a matter of fact, the SoftBank NAO robot is provided with a video camera with high resolution 1280x960 that can capture video with up to 30 frames per-second. Using the robot’s own camera will give a better perspective on the level of engagement between the children and the robot, since it is seen from the perspective of the robot. Also, it will give a better result for studying the eye gaze of the children to the robot.

The reviewer has some concerns about the generalization in using the word “Robot” in the title, while a robot "NAO" is used for the experiment, the look and feel of NAO is different from other robots that are bigger in size for example, ASIMO, Baxter, Pepper ..etc. NAO looks more like a toy-robot at least from size perspective, consequently, the children’s level of engagement during the learning experience might differ when a bigger “adult-sized” robot is used.

Minor issues:

In line 249 the authors used the notation “M=” and “SD=”, do you mean the mean and the standard deviation. Please write the complete wording of the abbreviation explicitly on the first use.

Also, in lines 256, 257 and 258 the authors used the notation “3;6” and “3;7” do you mean three years and six months, and three years and seven months, please verify!

In page 2 line 37, is it “visa versa” or “vice versa”, please check?

Reviewer 2 Report

The authors present a study that analyzes the influence of different types of feedback on the engagement and success of a learning task in a set of 3-to-4-years-old kids, and found little relation between robot and task engagement as well as how well the kids learned during the task. However, they did found evidence that eye-gaze is an important feature to use to measure engagement and that there is some influence of the presence of feedback from the robot in both the robot and task engagement over time.

The paper is clearly written, and only requires some minor issues to be fixed:

- The authors should state if there is any influence on the fact that all the tested kids were boys, and, thus, if their results may have a gender bias.

- In lines 252-253 it is stated that "Four children indicated that they wanted to stop participating during the experiment". Additionally, in lines 364-367 it is stated that "When children looked at the experimenters or asked them for help, the experimenters redirected the children’s attention back to the robot. When a child displayed signs of discomfort, experimenters comforted the child and tried to make him or her more relaxed." Although the authors do make a good case of why this happened (in short, 3-to-4-years-old kids have a very short attention span), they do also mention in lines 705-706: "There were large individual differences between children, some children did learn all the target words and some did not learn any words.", meaning, there were some kids that did benefit from the interaction with the robot. This reviewer proposes that the authors add additional analysis (with the data that the authors already have on hand) focusing only on kids that did benefit. This, to be clear, should be added so as to state that even though in average there is no observable learning benefits, for those that do benefit: what kind of benefit was observed?

- Figure 2 should be split into two figures, each being discussed in their respective sections (3.1.1 and 3.1.2.).

- Table 2 should be move to section 3.2.

- There is a small amount of typos/grammatical errors/style mishaps that this reviewer found which should be fixed:
. line 202: "Both our previous study as the study by Ahmad and colleagues" -> "Both our previous study and the study by Ahmad and colleagues"
. line 409: "had cues such as: when turning away from the robot" -> "had cues such as: turning away from the robot"
. line 596, 620: "figure 2" -> "Figure 2"
. line 627-631: this reviewer proposes the following change to this sentence:

"In a similar way that the absence of corrective feedback to help the children in the rest of the session might have reduced their attention for the learning task, possibly increased frustration [63] which consequently could lead to task disengagement."

->

"In a similar way that the absence of corrective feedback to help the children in the rest of the session might have reduced their attention for the learning task, it also possibly increased frustration [63] which consequently could have lead to task disengagement."
. line 719: "had to repeat the robot" -> "had to repeat the word to the robot"

Reviewer 3 Report

Great work! I really enjoyed reading your document.

Just a minor comment: Put the "0" to decimal numbers between -1 and 1, e.g., 0.23 instead of .23.
